# Sand Beach Nourishment: Experience from the Mediterranean Coast of Israel

**Menashe Bitan [1] and Dov Zviely [2],***

[1] Department of Maritime Civilizations, The Leon H. Charney School for Marine Sciences, University of Haifa, 199 Aba-Khoushi Avenue, Mount Carmel, Haifa 3498838, Israel; bitanmen@netvision.net.il

[2] Faculty of Marine Sciences, Ruppin Academic Center, Emek-Hefer 40250, Israel

\* Correspondence: dovz@ruppin.ac.il; Tel.: +972-52-5805-758

**Abstract:** Beach nourishment along the Mediterranean coast of Israel represents a new approach to mitigate coastal erosion by adding suitable sand to threatened beaches. This 'soft' solution has become more environmentally and economically acceptable than traditional 'hard' solutions, such as seawalls, revetments, detached breakwaters and groins. Beach nourishment projects have been implemented on the Israeli coast north of Ashdod Port (2011), north of Ashkelon Marina (2015) and in the south of Haifa Bay (2016–2017). The performance of these projects was analyzed and compared with nourishment projects along the Mediterranean beaches of Italy, France and Spain. Despite a lack of detailed documentation on most of the European nourishment projects, they proved more durable than the Israeli projects, which were compromised when the imported sand eventually washed offshore. Key factor for the Israeli projects' failure include the unsuitable morphology of the beaches; insufficient unit sand volume ($m^3/m$—volume of nourished sand per meter of the beach length); and imported sand that was too fine versus native sand. The unique physical conditions of the Israeli coast specifically, its open shelf and straight coastline subject to relatively high waves with a very long fetch—also contributed to the poor durability of the nourishment. To improve durability on future projects: imported grain size should be at least 1.5–2.0 times the native sand; unit sand volume should be 400–500 $m^3/m$; and supporting measures should be utilized as appropriate.

**Keywords:** coastal erosion; dredging; rainbowing; nourishment durability; unit sand volume

## 1. Introduction

Coastal erosion is a global phenomenon caused by the action of wind, waves, currents and sea-level changes, as well as by human intervention that accelerates the erosion rate [1]. For example, about 15,100 km of European coastline is retreating (out of a total of 101,000 km), and about 15 $km^2$ of land is lost each year [2]. Coastal erosion generates a significant threat to society and the economy in general, and to tourism in particular [3–6]. Some European beaches retreat up to several meters per year [7], and on the Mediterranean coasts of Italy, France and Spain the length of eroded beaches: 1500 km, 1200 km, 750 km, respectively, exceeds that of the stable beaches. In Greece, however, the 500 km of eroded beaches is less than the length of stable ones, as most of the coast is rocky [8].

To mitigate coastal erosion, 'hard' solutions, such as seawalls, revetments, detached breakwaters and groins, have been used since the Roman Imperial period until recently [1]. These coastal defense structures do not stop beach erosion, but transfer it with the longshore current [8]. Analysis demonstrates the inefficiency of 'hard' solutions in reducing erosion, and their negative impact on environmental quality. Such approaches have been used as emergency responses to problems without adequate knowledge of possible consequences [9,10].

In contrast, the application of 'soft' solutions, mainly sand beach nourishment, as well as dune reconstruction (or establishing new dunes) and beach dewatering, has become an alternative remedy for chronic sandy beach erosion [1,11,12]. Sand beach nourishment is typically a repetitive process that allows continued use of recreational beaches, and protects structures and built-up areas near the shoreline. It does not eliminate the physical factors that cause erosion, but mitigates their effects. This is the reason why beach nourishment is considered to be more environmentally acceptable, as many evaluations have shown its success in mitigating erosion caused by nature and/or human activity [3,13–16]. As a result, beach nourishment has become one of the most popular methods of coastal protection in the United States and Europe [8,14,17–21].

In 2011, sand beach nourishment was implemented for the first time on the Mediterranean coast of Israel. Since then this application has become the preferred method of the Israeli authorities for preserving and expanding eroded beaches. This paper analyses the physical conditions, design process and key factors that affected the results of sand beach nourishment projects in Israel (north of Ashdod Port, north Ashkelon, and south Haifa Bay) from 2011 to 2017 (Figure 1).

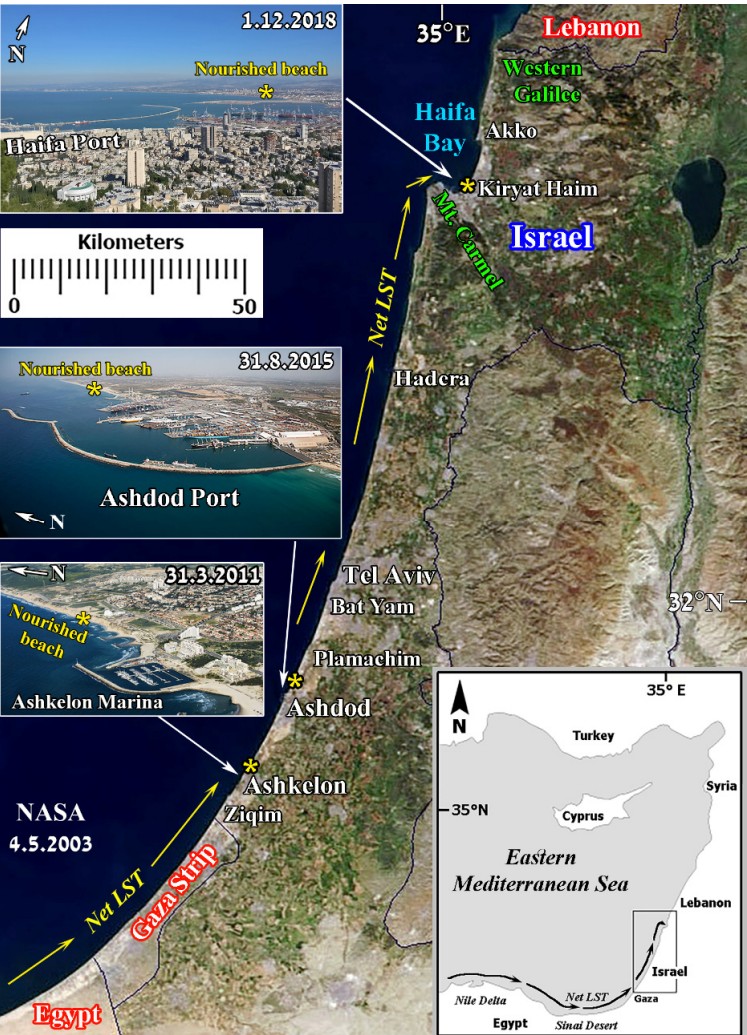

**Figure 1.** The Mediterranean coast of Israel and the three sites where beach sand nourishment was carried out between 2011 and 2017: south of Haifa Bay (top inset); north of Ashdod Port (central inset); north Ashkelon (bottom inset). Net longshore sand transport direction (yellow arrows). Background: MiddleEast.A2003031.0820.250m.jpg; Photographed by Descloitres, J., MODIS Rapid Response Team, NASA/GSFC, 31.1.2013.

In order to understand the performance of the Israeli projects better, they were compared with available information from beach nourishment projects carried out on the Mediterranean coasts of Italy, France and Spain. The small tidal range of the Mediterranean and its wave regime compared to oceans, allows this comparison, although far from perfect, for the Israeli coast.

## 2. Study Area

### 2.1. Physical Setting of the Israeli Coast

The Mediterranean coastline of Israel extends 195 km from the border of the Gaza Strip in the south to the Lebanese border in the north. It is generally a smooth coastline open to the west that gradually changes in orientation from northeast to almost north, with the exception of Haifa Bay, the Mount Carmel headland and a few small rocky promontories (Figure 1).

Most Israeli beaches are straight, flat and sandy, and less than 50 m wide [22]. Along coastal sections, such as Ashkelon, Palmachim, and between Bat Yam and Hadera, where the beach is backed by a calcareous sandstone (locally termed kurkar) cliff, the beach is generally less (and sometimes much less) than 30 m wide [23].

The Israeli coast and its inner shelf, from the shore to about 30 m water depth, can be divided into two main sedimentological provinces. The Southern Province stretches 175 km from Ziqim to the Akko promontory (northern Haifa Bay), and is considered the northern flank of the Nile littoral cell [24–26]. This region is mainly composed of fine quartz sand ($d_{50}$ = 125–250 μm) [27]. The sand from the Nile Delta is transported by longshore currents eastward to the northern coast of the Sinai, then north along the Israeli coast. These currents are generated by the radiation stress of breaking waves and shear stress of local winds. Wave-induced and wind-induced longshore currents occur in both directions. However, the long-term net longshore sand transport (LST) runs northward along the entire coast, up to Haifa Bay (Figure 1: bottom right inset) [26,28–30]. The Northern Province (the western Galilee coast) is a small, isolated and rocky littoral cell, partly covered with local coarse carbonate sand [23,31,32].

The Israeli Mediterranean wave climate can be divided into two seasons: summer (April to October) and winter (November to March). During the summer season, the wave climate is characterized by relatively calm sea with a wave height rarely exceeding 2 m (Hs < 2 m). In the winter season, however, the wave climate is characterized by alternating periods of calm seas and storm events of up to 5 m significant wave height (Hs) [33,34]. Since 1992–1993, high-quality directional wave data have been measured offshore Haifa and Ashdod (110 km apart) by the Coastal and Marine Engineering Research Institute (CAMERI) on behalf of the Israel Ports Company (IPC). At these sites, where water depth is about 24 m, a Datawell directional wave-rider buoy is deployed to acquire 30-min records of surface elevation and directional spectral information [30]. A study based on long-term statistical analysis shows that about 6% of waves recorded in Haifa between 1 April 1994 and 31 March 2004 were higher than 2 m (Hs > 2 m), and only 1.4% of wave heights were 3.0–4.5 m [35].

By using the Weibull distribution with a 3.7 m *Hs* threshold, an analysis of extreme wave events recorded in Ashdod during the period of 1 April 1992–31 March 2015 shows that the significant wave heights with 10, 20, 50 and 100-year return period are about 6.54 m, 7.07 m, 7.75 m and 8.27 m respectively [36]. The analysis of storm events recorded at Haifa during the period of 25 November 1993–31 March 2015, shows that the average number of storms (Hs > 3.5) per year is about five [34], and during the last 25 years four major storms with Hs > 7 m occurred in February 2001, December 2002, December 2010 and February 2015. These events show that the Israeli coast is affected by relatively high waves.

Based on high-quality wave measurements recorded in Haifa and Ashdod between 1993 and 2016, the closure depth (h) for the Israeli coast was calculated according to [37] Birkemeier (1985) expression: h = 1.57 $H_e$, where $H_e$ is the nearshore significant wave height (m) exceeded only for 12 h per year. The calculated $H_e$ for the Israeli coast, except Haifa Bay, is 5.1 to 5.8 m which means the closure depth

is between 80. and 9.1 m [22]. In Haifa Bay however, the calculated $H_e$ is 3.0 to 3.3 m which means the closure depth is between 4.7 and 5.2 m [38].

*2.2. Coastal Sections: Physical Description and Nourishment Projects*

Detailed data gathered from reports, maps, aerial photographs, grain-size analysis and field observations were used to study the physical conditions of each coastal section nourished between 2011 and 2017, and analyze the design process and key factors that affected the project performance.

2.2.1. North of Ashdod Port

Ashdod Port is situated on the southern coast of Israel about 30 km south of Tel Aviv (Figure 1). It handles the largest cargo volume, and is the major gateway for cargo to and from the State of Israel. The port was built on a straight sandy beach backed by sand dunes between 1961 and 1964, and started operations in 1965. When the port was completed its main breakwater length was 2200 m, and that of the lee breakwater was 900 m. At that time, the head of the main breakwater was located at a water depth of 15 m, projecting about 1100 m from the shore [39]. Between 2001 and 2005, the main breakwater was extended by 1150 m to protect a new large container terminal (Eitan Port). The main breakwater head was located at a water depth of 21 m, some 1600 m from the shoreline (Figure 2). Since 2015, another huge container terminal (Southport Terminal) to handle the largest container vessels (Class EEE) has been under construction, and the terminal and the additional main breakwater extension are set to become operational in 2021.

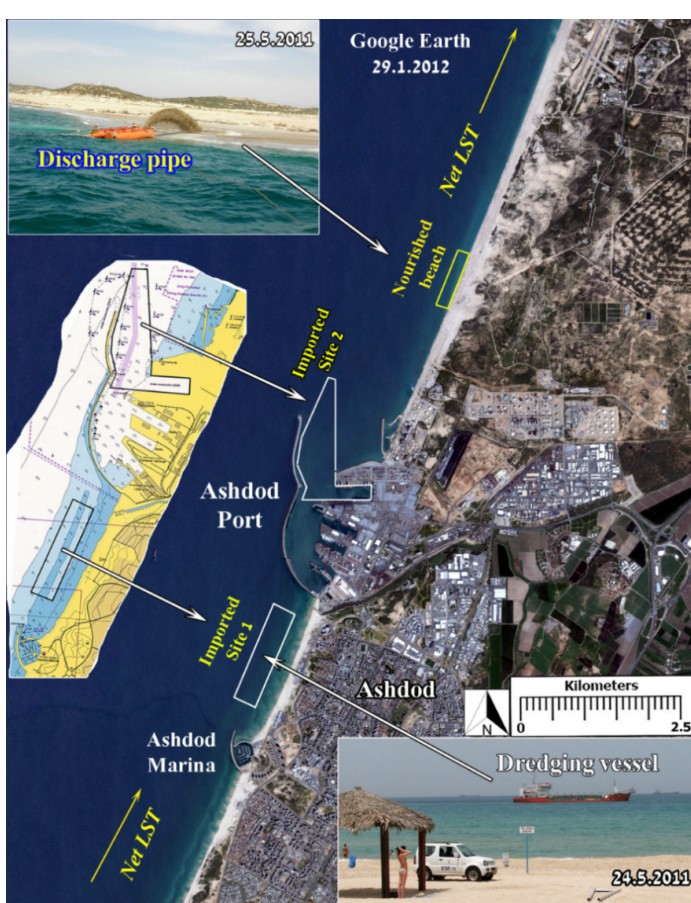

**Figure 2.** North of Ashdod Port sand beach nourishment project (May–August 2011). Dredging vessel operation (bottom inset); imported sand sites superimposed on local nautical chart (central inset); nourishment rainbowing operation via discharge pipe (top inset). Background: Google Earth image, 29 January 2012.

Ashdod Port is the largest obstacle blocking the northward LST regime along the southern Israeli coast, and its main breakwater serves as a huge sediment trap [39,40]. An assessment based on a comparison of bathymetric maps shows that a volume of sand of about 7.6 million m$^3$ was deposited south of the main breakwater between 1965 and 2013 (about 158,000 m$^3$ yearly on average). As a result, the shore stretching to a distance of 3.5 km south of the port was widened by 40 m on average, and more than 100 m near the main breakwater, between 1964 and 2010 [41]. North of the port, however, the comparison maps show a severe erosion of about 5 million m$^3$ between 1965 and 2013 (about 104,000 m$^3$ yearly on average). In this context, the sand from the coast north of the port was extensively exploited for building purposes prior to its construction, and when the port was built it was already eroded and rocky [39]. Thus, this coast could not have recovered naturally since 1965.

The first sand beach nourishment in Israel was carried out north of Ashdod Port between May and August 2011. The aims of this project were: (1) to bypass sand from the huge sandbar stretching south of Ashdod Port main breakwater and (2) to nourish the eroded coast north of the port. For the nourishment, a total volume of sand of about 315,000 m$^3$ was dredged from two sites: between the Ashdod Marina and Ashdod Port at a water depth of 5 to 8 m (~100,000 m$^3$) (Figure 2: Imported Site 1), and in Ashdod Port area (~215,000 m$^3$) (Figure 2: Imported Site 2). The sand was deposited between the coastline and water depth of 3 m by rainbowing via a discharge pipe at the bow of the dredging vessel (SIMI—operated by EDT Marine Construction) anchored at a water depth of 6 m. The dredging vessel conducted up to four cycles per day, about 800–1600 m$^3$ of sand per load. At the end of the operation, a 1 km-long coastal section had received nourishment, starting about 2.8 km north of the Ashdod port's lee breakwater in an area of 30 by 80 m (Figure 3a). No grain size analyses of the imported site and nourished beach were made pre-nourishment. A previous study showed that mean grain size ($d_{50}$) in the imported site was in the range of 170–180 μm (fine sand) [42], while the planned nourished coast contained medium to coarse sand [43] (Appendix A: Table A1).

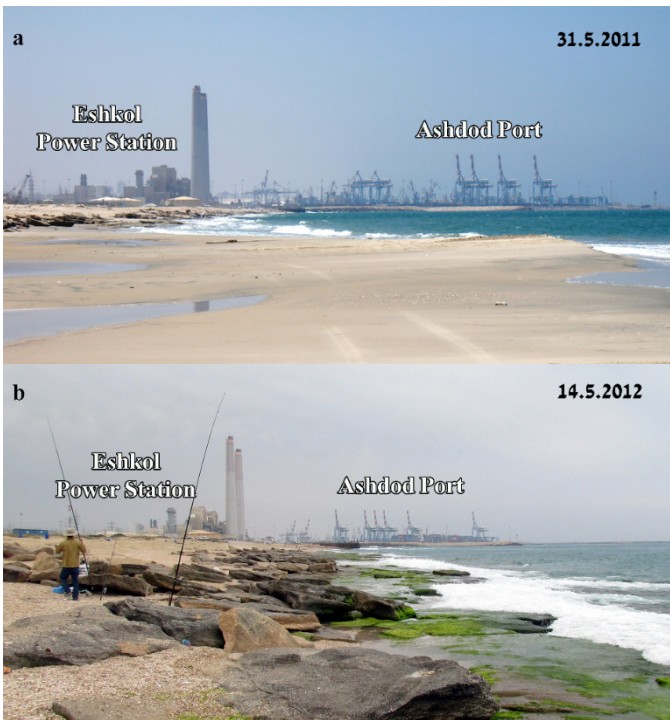

**Figure 3.** The rocky (beachrock) coast north of Ashdod Port: (**a**) during the nourishment operation (May–August 2011); (**b**) about a year later (May 2012).

In spring 2012, a few months after the nourishment was completed, a site visit found no evidence of the massive sand nourishment, while the beach had reverted to its previous rocky state (Figure 3b).

A comparative analysis of bathymetric maps showed that in July 2012 half of the nourished sand volume had left the nourished site, and the rest had migrated to deeper water [44].

### 2.2.2. North Ashkelon

Ashkelon is the southernmost city on the Mediterranean coast of Israel. The city has a 12 km stretch of beautiful sandy beaches which attract tourists from Israel and abroad. Ashkelon Marina is situated on the central coast of Ashkelon (Figure 4), about 50 km south of Tel Aviv, and about 11 km north of the border of the Gaza Strip (Figure 1). The Marina was built between 1992 and 1994 on a straight sandy beach backed by an 18 m-high kurkar cliff, and about 200 m north of the central bathing beach of Ashkelon (Delilah Beach). This beach was significantly widened after the construction of three detached breakwaters in 1984. The length of the Marina's main breakwater is 720 m, and that of the lee breakwater is 285 m. The head of the main breakwater is located at a water depth of about 6 m, and projects about 350 m into the sea (Figure 4).

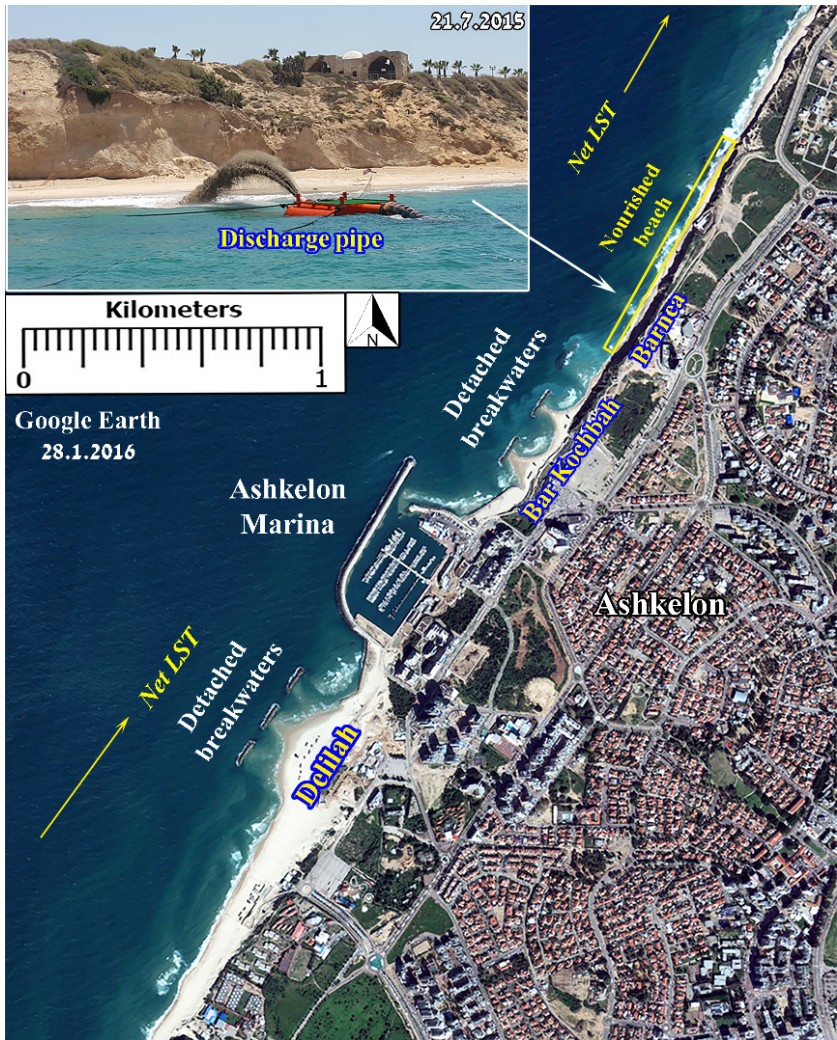

**Figure 4.** North Ashkelon sand beach nourishment project (July–October 2015). Nourishment rainbowing operation via discharge pipe (top inset). Background: Google Earth image, 28 January 2016.

To mitigate the expected coastal erosion that would develop north of the planned Marina as a result of its interruption to the northward wave-induced LST, a series of three detached 100 m-long breakwaters were built at a water depth of 3 m about 100 m from the shoreline. Subsequently a wide tombolo beach (Bar Kochba Beach) developed behind the breakwaters, but the erosion shifted

northward to Barnea Beach and further north (Figure 4). A comparative analysis of aerial photographs shows that the shoreline up to 1 km north of the detached breakwaters had retreated by 32–56 m between 1986 and 2014 [45]. This resulted in the collapse of the unstable coastal cliff, which retreated about 30 m mainly after the marine construction, leaving the front of the Harlington Hotel (former Holiday Inn Hotel) about 30 m from the cliff edge [41].

The retreat of the cliff has also exposed centuries-old archaeologically valuable remains buried in the poorly cemented layers of the cliff. Stabilization of the Barnea Beach has become urgent in order to protect further collapse of the cliff.

Sand beach nourishment projects in north Ashkelon were carried out in 2015 and 2016 to expand Barnea Beach in order to provide temporary protection from wave action at the foot of the cliff. The first nourishment was carried out between 9 July 2015 and 10 August 2015, when 71,200 m$^3$ of sand was dredged from the Ashkelon (Rutenberg) power station cooling water basin and deposited along a 750 m length of coast, starting about 150 m north of the northern detached breakwater.

The sand was deposited between the shoreline and water depth of 2 m by the method used in Ashdod in 2011, and was bulldozed ashore (Figures 4 and 5).

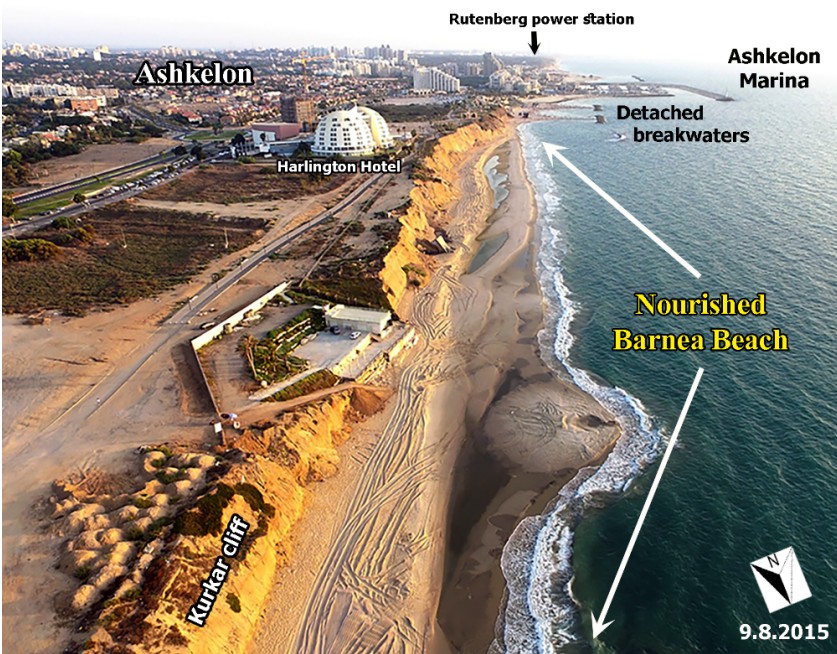

**Figure 5.** Aerial photograph of the coast north of Marina Ashkelon at the completion of sand nourishment (9 August 2015).

A grain size analysis of Barnea and Bar Kochba beaches was conducted pre-nourishment by Geo-Prospect Ltd. (Jerusalem, Israel), for the Mediterranean Coastal Cliffs Preservation Government Company Ltd. (MCCP) (Netanya, Israel). The result was compared to grain size analysis of the supply (imported) sand conducted by the Israel Electric Cooperation in 2013. The comparison shows that the mean grain size in the planned nourished beaches was about 250 μm, while in the imported site it was about 200 μm (Appendix A: Table A1).

The second and third nourishments were carried out in February and December 2016, when sand volumes of 11,000 m$^3$ and 7800 m$^3$ respectively were dredged from Ashkelon Marina and deposited north of the detached breakwaters.

A comparison between topographic and bathymetric maps shows that prior to the first nourishment (13 June 2015) the beach profile slope from the coastline to a distance of 200 m offshore was moderate. Three weeks after nourishment (28 August 2015) the beach was wider by 30 m, and its slope had become steeper. Six months later (22 April 2016) the beach had retreated to its initial width, and nourished sand

had drifted 50 to 200 m offshore, and was scattered in a water depth of 1 to 6 m. These morphological changes clearly show the impact of winter storms. Concerning the continuing cliff retreat, monitoring along 3 km north of the Marina by airborne light detection and ranging (LIDAR) between winter 2016 and winter 2017 revealed 69 collapse events and about 2150 m³ of the cliff material washed offshore in spite of the nourishment [46].

### 2.2.3. South Haifa Bay

Haifa Bay, in northern Israel, is the most significant morphological feature on the southeastern Mediterranean coast. It is open to the west, bordered by the Carmel headland to the south and Akko promontory to the north (Figure 1) [35]. The bay's 18 km-long coastline is crescent-shaped, with about 6 km of continuous marine structures (including Haifa Port, Kishon Harbor, Haifa power plant cooling basin and seawalls) in the southern part (Figure 6), and about 12 km of sandy beaches in the eastern part. Haifa Port is situated about 110 km north of Ashdod Port, and about 30 km south of the Lebanese border. It is Israel's largest and leading container port, and includes facilities allowing for shipping and transportation of all types of cargo, as well as docking facilities for large passenger liners. The port was built between 1929 and 1933 on the seafront of the city of Haifa, and on completion the length of its main breakwater was 2210 m, and that of the lee breakwater was 765 m [35]. At that time, the head of the main breakwater was located at a water depth of 11.5 m, projecting about 1150 m from the shore. Between 1978 and 1980 a container terminal (the Eastern Quay) was built in the eastern part of the port. To protect this terminal from waves from the northwest the port's main breakwater was extended by 600 m to a water depth of 13.5 m. During 2005 and 2008 another container terminal (the Carmel Terminal) was built in the eastern part of the port, between the Eastern Quay and the power plant cooling basin (Figure 6). The construction of the latest container terminal (Bayport Terminal) has been ongoing since 2015, and it is due to become operational during 2021. This huge terminal, designed to handle Class EEE container vessels, is located relatively close to the Kiryat Haim beaches. To protect the Bayport Terminal, the main breakwater was extended by 882 m to a total length of 3682 m, and its head located at a water depth of about 20 m.

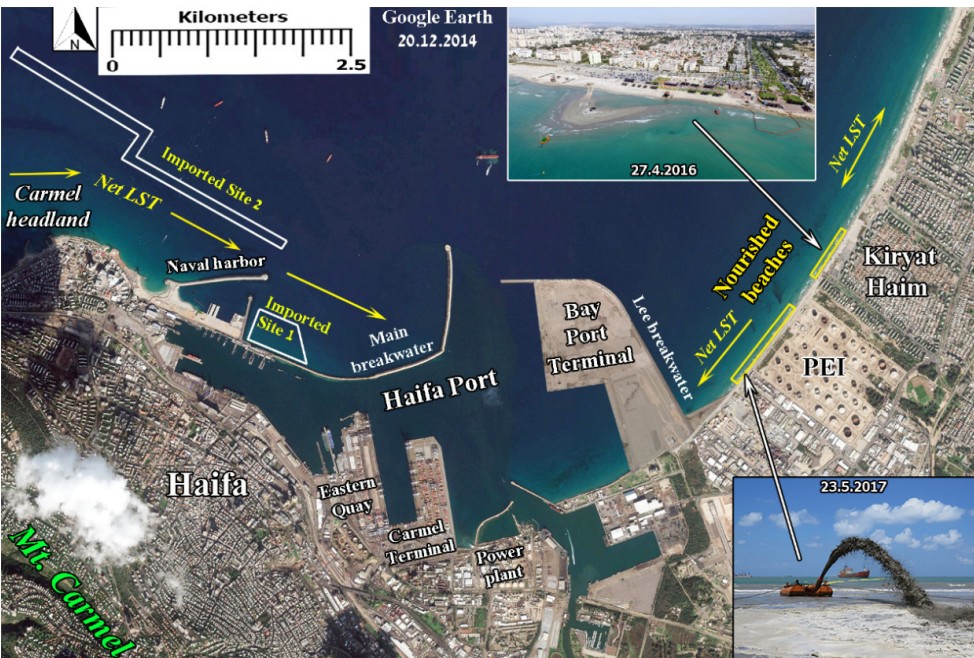

**Figure 6.** South Haifa Bay sand beach nourishment projects (May 2016–June 2017). Aerial photograph of Kiryat Haim bathing beach (Yehudit Naot Beach) during the first nourishment operation (**top** inset). Nourishment rainbowing operation via discharge pipe south of Petroleum and Energy Infrastructure Ltd. (PEI) (23 May 2017) (**bottom** inset). Background: Google Earth image, 20 December 2014.

For the last 7900–8500 years, Haifa Bay area has been the northernmost final depositional sink of the Nile littoral cell [26,35,47]. During this period sand bypassing the Carmel headland from south to north was transported unimpeded to the shore of the Bay by longshore currents [25]. The construction of Haifa Port (1929–1933) in the southern part of the Bay created a large trap for migrating sand along its main breakwater. The total amount of sand trapped between 1929 and 2004 has been estimated at about 5 million m$^3$, or an average of 66,000 m$^3$/year. Only a small amount of sand (8000–10,000 m$^3$/year) bypassed the main breakwater head during this period, and drifted eastwards to the shore of the Bay [26]. A study shows that between 1799 and 1928 (construction of the previous ports), the Bay's coast expanded by 50 to 150 m (averages of 0.4–1.2 m/year). However, between 1928 and 2006, most of the coast was in a steady state, with seasonal fluctuations of less than ± 20 m [35,38]. In the last decade, severe erosion of more than 20 m has developed in several beaches in the south of Haifa Bay. The reasons for this negative change are not fully understood, but it seems that the key factors are the latest extension of the Haifa Port main breakwater and the construction of the Bayport lee breakwater.

To mitigate coastal erosion in the southern part of Haifa Bay, especially along the beach in front of the Petroleum and Energy Infrastructure Ltd. (PEI) fence (Figures 6 and 7) and Kiryat Haim bathing beaches, several beach nourishment activities were carried out during 2016 and 2017. In order to expand the eroded beaches, the sand was deposited between the shoreline and water depth of 2 m by the method used in Ashkelon in 2015.

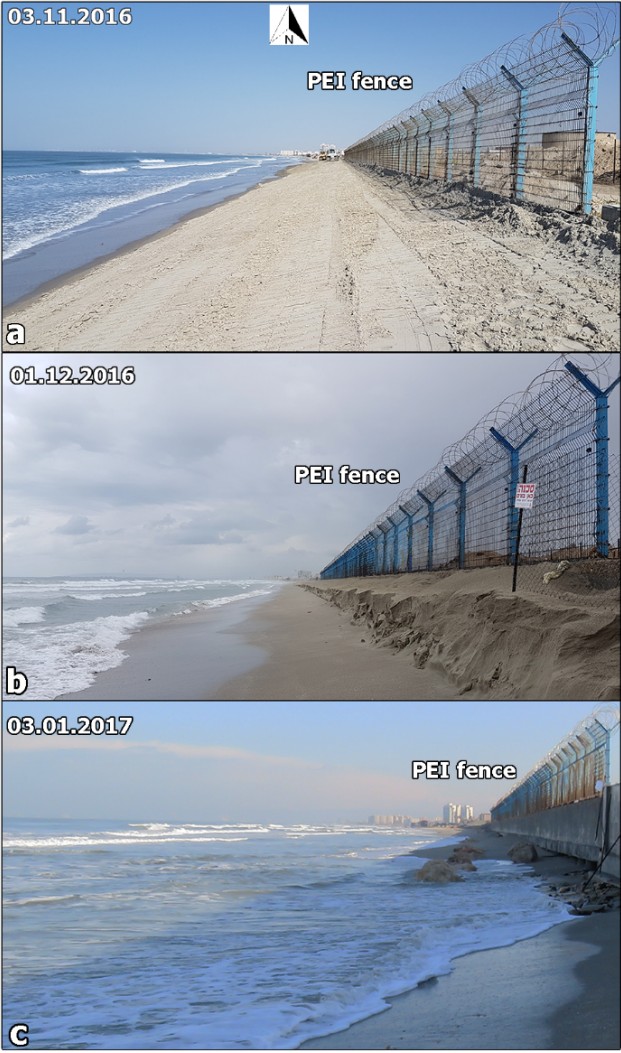

**Figure 7.** The sandy beach along the southern part of PEI fence: (**a**) after nourishment; (**b**) one month later; (**c**) two months after nourishment.

The first nourishment was carried out between April and May 2016, when a sand volume of 70,000 m$^3$ was dredged near the outer side of the main breakwater of Haifa Port (Figure 6: Imported Site 1) and deposited in two sites: (1) a coastal section of 450 m along the Kiryat Haim bathing beaches (45,000 m$^3$); and (2) a coastal section of 250 m along the southern part of the PEI fence (25,000 m$^3$). No grain size analyses of the imported site and nourished beaches were carried out before the nourishment. A previous study, however, supplied some data, showing that the mean grain size in the imported site was in the range of 160–200 μm, while in the nourished coasts the mean grain size was in the range of 149–210 μm [35] (Appendix A: Table A1).

The second nourishment was carried out in October 2016, when a sand volume of 100,000 m$^3$ was dredged north of the main breakwater of the new naval harbor (Figure 6: Imported Site 2), and nourished a coastal section of 500 m along the southern and central parts of the PEI fence. The nourished sand was bulldozed to decrease sand porosity. A pre-nourishment grain size analysis of six samples in the imported site showed that the mean grain size was in the range of 140–170 μm, which was compatible with the sand in the nourished coast. When the nourishment ended the 3 m high PEI fence was protected by a triangular sandbar that widened the beach by more than 20 m (Figure 7a). However, a few days after the nourishment, the sandbar began to erode due to a typical seasonal storm, and in less than three weeks most of the sandbar had almost disappeared (Figure 7b). At the beginning of December 2016 an extreme storm with a significant wave height (*Hs*) of 5.2 m and a maximum wave height (Hmax) of 10 m destroyed all evidence of the nourishment (Figure 7c).

In order to maintain a dry path for walking along the whole PEI fence, another nourishment was carried out between May and June 2017, when a sand volume of 185,000 m$^3$ was dredged north of the main breakwater of the new naval harbor, and nourished a coastal section of 600 m along the southern and central parts of the fence. A grain size analysis of six sand samples in the imported site showed that the mean grain size was in the range of 120–140 μm, which was finer than that in the nourished coast. When the nourishment ended, the coast had been widened by 40 m on average. This lasted until the fall, and was totally eroded by storm waves by mid-January 2018, seven months after nourishment. In the past year, the erosion has continued to develop, and now the sea has flooded a 300 m section along the southern part of the PEI fence.

## 3. Discussion

Preserving bathing beaches is the most common reason for sand beach nourishment in EU Mediterranean countries, as they are used for recreational activity [8,9,14,17,48–54]. The physical conditions and key factors of successful nourishment projects conducted on the Mediterranean coasts of Italy, France and Spain have been analyzed to better understand the sand beach nourishment performance carried out in Israel between 2011 and 2017 (see Sections 2.2.1–2.2.3, Figure 8). In most projects in Italy, France and Spain, the aims of beach nourishment were to maintain and widen recreational beaches, protect coastal constructions, infrastructure and cliffs from wave action, and prevent sea flooding [3,6,14,17,55–58]. In addition to project aims, quantitative objectives for the nourished beaches, such as durability, width of beach, refill period and reservoir availability were defined before nourishment. In Israel, the project aim was to widen eroded sandy beaches: however, no quantitative objectives were defined.

### 3.1. Physical Conditions

An important consideration of beach nourishment is the type of coastal morphology and its wave regime (e.g., straight and smooth beaches facing a long fetch, or bays and pocket beaches that have relatively good protection from waves) [14,16,59]. Wave characteristics (wave height, period and direction) and storm surge level affect the tendency to beach erosion [17,60,61].

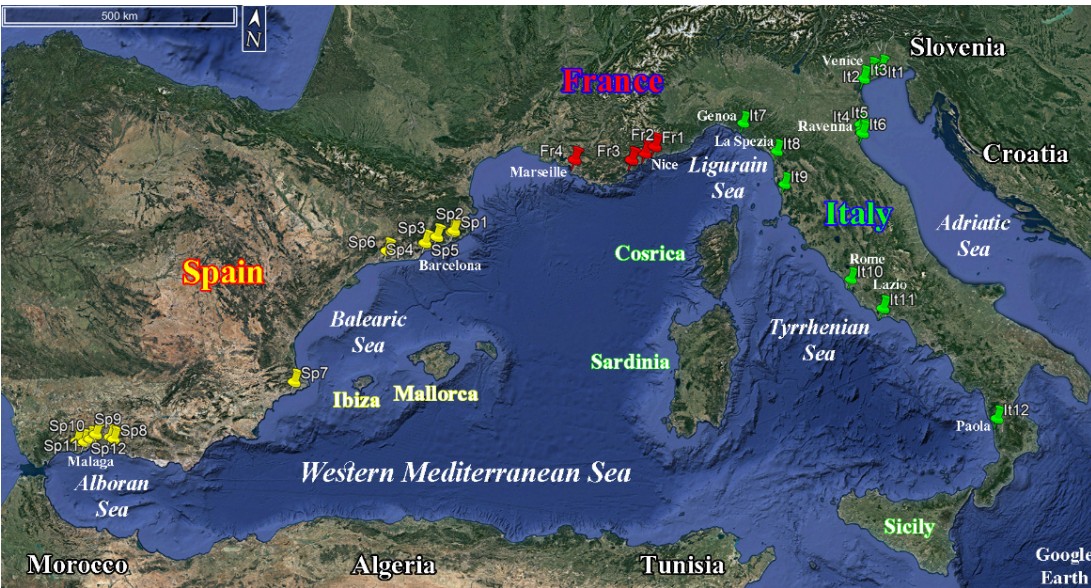

**Figure 8.** Selected beach nourishment sites along the Mediterranean coast of Spain (Sp1–Sp12), Italy (It1–It12) and France (Fr1–Fr4) (see Appendix A: Table A1).

The Israeli coast is an open shelf straight coastline affected by relatively high waves with a very long fetch (up to about 2400 km) [34,62], while the coasts of most European Union (EU) Mediterranean countries are relatively protected (e.g., bays, pocket beaches) from storm waves [63–65].

### 3.2. Native and Imported Sand Grain Size

For successful nourishment, compatibility of the imported sand with the native beach sediment is vital [61,66]. Sediment that is somewhat coarser than native material enhances the longevity of the nourished beach. Imported material of grain size finer than that of the native material will be easily eroded by wave action and coastal currents; and a different composition will cause environmental harm [8,11,13,14,16–18,51,67–69].

The relation of imported to native grain size used for nourishment projects in Italy and Spain (no data for France) was 1.2–10.0 and 1.5–5.0 times coarser than native sand, respectively. However, documentation for grain size distribution analysis was not presented in all the projects. Such data could provide a stability index to determine the proper sand volume for effective nourishment [70]. It can also indicate the preferred sediment type (e.g., sand, gravel or shingle) suitable for the objective, such as the experience with gravel beach nourishment of the Marina di Pisa Beach in Italy [12]. Data from the Atlantic nourishment projects show that the ratio grain size of imported to native sand was 1.00–1.86:1 in Germany, 1.08–1.50:1 in Holland and 2.50:1 in France [17]. In most projects in Israel, however, the borrowed sand was finer than native sand, or nearly compatible (i.e., a ratio of 0.80:1 in Ashkelon and 0.90:1 in Haifa Bay), i.e., much finer than in the Mediterranean and Atlantic European projects.

### 3.3. Durability

Durability represents the longevity of the nourishing project that has a crucial impact on the economic success of the project [71]. Average durability is defined as the time until half of the nourished material remains [72]. It can also be measured by the first refill period after the end of the initial nourishment, or by the percentage per year of sand left on the nourished site [17]. The very sparse documentation of this significant factor on the Mediterranean coasts of Italy, France and Spain indicates a longevity of 1.5 to 8.0 years, or quite near to equilibrium (Appendix A: Table A1). Documentation of Atlantic Europe projects, however, shows much higher durability rates of 5–33% after 6 years in Germany, 8–15 years in Holland, 80–100% after 10 years in England, and 5–15 years in Denmark [17].

In Israel, however, the project durability after nourishment was poor, with a very short lifetime, and average erosion rates of 1270 m$^3$ per day in Ashdod, 252 m$^3$ per day in Ashkelon, and 507 m$^3$ per day in Haifa Bay. The average erosion rate in Haifa Bay after the November 2016 nourishment was 3300 m$^3$ per day.

### 3.4. Volume of Nourishment Sediment

The unit volume of sediment involved in the nourishment process is expressed as m$^3$/m of beach length. This measure is crucial for the success of the nourishment project, and usually a higher unit volume rate promises better durability of the nourished beach [13,14,17,52,68,73–75]. It does not dictate the sediment volume needed pre-nourishment, but is the result of experience of many nourishment projects carried out around the Mediterranean, where unit volumes were up to 500 m$^3$/m in Italy, 400 m$^3$/m in France, and even an extreme rate of 1700 m$^3$/m in Spain [17]. More data from the Atlantic European nourishment projects is available, although not complete. For example: the average unit volume is 385 m$^3$/m in Germany, 733 m$^3$/m in Holland, 570 m$^3$/m in England, and 230 m$^3$/m in Denmark. The projects in Israel, however, used a much smaller sand volume (209 m$^3$/m in Ashdod, 95 m$^3$/m in Ashkelon and 200 m$^3$/m in Haifa Bay).

### 3.5. Supporting Coastal Structures

In some cases, sand beach nourishment can be supported by detached breakwaters and groins in an effort to decrease the amount of nourished sand and increase the 'lifetime' (i.e., durability) of the nourishment project [8,9,17,76]. Most Italian projects favored supporting measures, and in the French and Spanish projects, although no supporting measures were planned, coastal structures already existed. In the Israeli projects, however, no supporting measures were planned.

## 4. Conclusions

(1) The first steps of Israel towards a soft solution for coastal erosion were the unsuccessful beach sand nourishment projects implemented in north of Ashdod Port, north Ashkelon, and in south Haifa Bay between 2011 and 2017. The short-term durability of these projects was compromised by failure to consider the unique physical characteristics of the Israeli coast by use of an insufficient volume of too-fine imported sand. However, these failures can provide experience for future success.

(2) Based on successful projects in the EU Mediterranean countries and the physical conditions along the Israeli coast, essential supporting measures such as coastal constructions can improve the durability of beach sand nourishment on the Israeli coast.

(3) A minimum target volume of nourished sand per meter of the beach length should be determined. Based on the experience of EU Mediterranean countries and other relatively moderate wave regime coasts with low tides, a unit volume of about 450 ± 50 m$^3$/m is needed for certain success of beach sand nourishment.

(4) The nourished sand should reflect the project objectives, and imported grain size should be at least 1.5–2.0 times the native sand.

(5) Successive bathymetric survey, granulometric analysis and fluorescent sand tracing are needed to understand of the dynamics of the nourished sand in the littoral zone better.

(6) It is recommended to start beach nourishment at the end of the winter storm season (April–May) in order to gain more benefit from the project. However, wintertime is sometimes preferable, since waves can clean the sediment and reshape the beach profile naturally.

(7) In places where a sea wall is located on the backshore, it is recommended to nourish the sand in the foreshore rather than to deposit it on the beach.

(8)   Detailed documentation of every nourishment project, such as native and imported grain size, m³/m unit volume, durability, etc., is essential for future nourishment design, and should be a condition for nourishment financing.

**Author Contributions:** Conceptualization, M.B. and D.Z.; investigation and resources, M.B.; writing—original draft preparation, M.B.; writing—review and editing, D.Z.; visualization, D.Z. supervision, D.Z.; All authors have read and agreed to the published version of the manuscript.

**Funding:** This research received no external funding.

**Conflicts of Interest:** The authors declare no conflict of interest.

## Appendix A

**Table A1.** Selected beach nourishment sites along the Mediterranean coast of Spain (Sp1–Sp12), France (Fr1–Fr4), Italy (It1–It12) (Figure 8), and Israel (Figure 1): Physical characteristics and main nourishment factors. (n/d: no data).

| Beach Name (Site) [Ref] | Coordinates (Lat., Long.) | Beach Type | Unit Volume m³/m | Native Sand Grain Size (µm) | Imported Sand Grain Size (µm) | Nourishing Season | Fill Type | Supporting Measures |
|---|---|---|---|---|---|---|---|---|
| Santa Cristina (Sp1) [48] | 41°41′30″ N 02°49′15″ E | Pocket beach | 450 | Fine (150) | Medium | 2009 before tourism season | Onshore | Rubble mound |
| S'Abanell Beach (Sp2) [77] | 41°40′49″ N 02°47′15″ E | Semi-enclosed | 446 | Very coarse (1,200) | n/d | 12/2007; 5/2008; 9/2009 | Onshore | None |
| Maresme Coast Barcelona (Sp3) [78] | 41°31′01″ N 02°10′01″ E | Quite flat | 97 | n/d | Coarse sand | Before tourism season | Onshore | 11 Groins |
| Bogatell Beach (Sp4) [20] | 41°19′48″ N 02°12′43″ E | Between two jetties | 120 | n/a | Coarse sand (450–900) | Before tourism season | Onshore | None |
| La Barcelonnette (Sp5) [20] | 41°13′25″ N 02°15′21″ E | Between two jetties | 113 | n/a | Coarse sand (450–900) | Before tourism season | Onshore | None |
| Altafulla Beach (Sp6) [60] | 41°08′03″ N 01°22′03″ E | Half-opened, between two capes | 69 | Fine (120–200) | Coarse sand (600) | Last 1990-beginning 1991 | n/a | Detached breakwater |
| Poniente Beach (Sp7) [17] | 38°34′41″ N 00°09′49″ W | Embayed | 318 | n/d | n/d | Before tourism season | Berm | n/d |
| Torrox Beach (Sp8) [17] | 36°43′46″ N 03°57′56″ W | Embayment headland | 280 | n/d | n/d | Before tourism season | Berm | n/d |
| Algarrobo Beach (Sp9) [17] | 36°08′14″ N 04°20′31″ W | Half-opened, between two capes | 241 | n/d | n/d | Before tourism season | Berm | n/d |
| Malagueta Beach (Sp10) [17] | 36°43′10″ N 04°24′09″ W | Embayed | 680 | n/d | Coarse | Before tourism season | Berm | Detached breakwater |
| Carihuela Beach (Sp11) [17] | 36°36′28″ N 04°30′17″ W | Quite flat | 277 | n/d | n/d | Before tourism season | Berm | n/d |
| Fuengirola Beach (Sp12) [17] | 36°32′19″ N 04°37′23″ W | Embayed | 1714 | n/d | n/d | Before tourism season | Berm | n/d |
| Marbella Beach (Sp13) [17] | 36°30′34″ N 04°52′51″ W | Embayed | 239 | n/d | n/d | Before tourism season | Berm | n/d |
| Eraclea (It1) [79] | 45°32′49″ N 12°46′08″ E | Internal sea. Large bay (100 km) wide | 941 | Fine (200) | As native (well sorted) | Spring 1994 | berm | 32 groins and beach grass |
| Cavallino (It2) [79] | 45°28′12″ N 12°29′45″ E | Internal sea. Large bay (100 km) wide | 181 | Fine (150) | Slightly coarser than native | n/a | berm | 32 groins and beach grass |
| Isola di Pellestrina (It3) [79] | 45°17′34″ N 12°18′58″ E | Internal sea. Large bay (100 km) wide | 418 | Fine (190) | Medium (220) | n/d | berm | 18 groins and trees |

**Table A1.** *Cont.*

| Beach Name (Site) [Ref] | Coordinates (Lat., Long.) | Beach Type | Unit Volume m³/m | Native Sand Grain Size (μm) | Imported Sand Grain Size (μm) | Nourishing Season | Fill Type | Supporting Measures |
|---|---|---|---|---|---|---|---|---|
| Porto Corsini Casal Borsetti (It4) [17] | 44°34′01″ N 12°16′40″ E | Quite flat beach | 140 | n/d | n/d | n/d | n/d | 10 groins and submerged barrier |
| Lido Dante–Marina di Ravenna (It5) [17] | 44°24′26″ N 12°18′43″ E | Quite flat beach | 200 | n/d | n/d | n/d | n/d | 17 groins |
| Foce Fiume Savio (It6) [17] | 44°19′53″ N 12°20′40″ E | Quite flat | 167 | n/d | n/d | n/d | n/d | 25 groins |
| Lavagna (It7) [17] | 44°18′29″ N 09°20′51″ E | | 160 | n/d | n/d | n/d | n/d | 2 groins |
| Porto Canaledi Viareggio (It8) [17] | 43°53′38″ N 10°14′56″ E | Bay | 267 | n/d | n/d | n/d | n/d | n/d |
| Cecina Mare (It9) [17] | 43°25′54″ N 09°39′17″ E | Bay | 108 | n/d | n/d | n/d | n/d | 7 groins and submerged barrier |
| Lido di Ostia (It10) [17] | 41°46′46″ N 12°16′08″ E | Quite flat, low fetch | 467 | n/d | n/d | n/d | n/d | Submerged barrier |
| S. Felice Circeo (It11) [17] | 41°15′34″ N 13°05′28″ E | Embayed, low fetch | 400 | n/d | n/d | n/d | n/d | Submerged barrier |
| Paola S. Lucido (It12) [17] | 39°20′06″ N 16°02′10″ E | Quite flat, low fetch | 193 | n/d | Medium (350) | n/d | n/d | 18 groins and submerged barrier |
| Nice, Baie des Anges (Fr1) [80] | 43°39′24″ N 07°12′52″ E | Beach between two headlands | 124 | n/d | Very coarse gravel (5–10 cm) | Before bathing season | Onshore | n/d |
| La Croisette (Fr2) [17] | 43°33′36″ N 07°02′19″ E | Pocket beach | 120 | Fine (200) | n/d | n/d | Onshore | Three groins |
| Fréjus-Saint Aygulf (Fr3) [17] | 43°23′37″ N 06°42′42″ E | Bay | 400 | n/d | n/d | n/d | Onshore | Three breakwaters |
| Le Prado (Fr4) [17] | 43°17′05″ N 05°22′28″ E | Bay | 115 | n/d | n/d | n/d | Onshore | Three breakwaters |
| North Ashkelon (Israel) | 34°33′51″ N 31°41′23″ E | Quite flat | 95 | 250 | 200 | End of bathing season | Onshore | noun |
| North of Ashdod Port (Israel) | 34°39′35″ N 31°51′2.0″ E | Quite flat | 209 | Medium to coarse | 170–180 | May–August | Onshore | noun |
| South Haifa Bay (Israel) | 34°33′51″ N 32°49′21″ E | Quite flat | 200 | 149–210 (2016) 140–170 (2017) | 160–200 (2016) 120–140 (2017) | End of bathing season (2016) Spring–Summer(2017) | Onshore | noun |

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
