# Peer review of "Sand Beach Nourishment: Experience from the Mediterranean Coast of Israel"

_jmse, doi:10.3390/jmse8040273_

Round 1

Reviewer 1 Report

This is essentially a brief documentation on the three projects along the Israeli coast. Given the grain-size of the borrow sand, it's not surprising that the fill had a short lifetime. The authors seem to attribute the problem to a disruption of the LST from the south. there seem to be substantial interruptions farther south, but in the vicinity of these project I see little or no evidence of regional northward drift or interruption at these site. In addition, the discription of some of the loses seem to be cross-shore rather that longshore transport.  The problem seems to be, at least in part, due to the compartmentalization of the shoreline by the local structures. Was hydrodynamic or wave modeling done?

line 87. The beach profiles probably vary greatly around the structures, but is there any estimate of a clousure dept?

line 95. What is the rate of LST?  Any evidence of divergence along the coast? 

Figure 3. Could cliff erosion be (or have been)  the source of sand to the beach?

line 160 and 282. were the repeated renourishments part of the original plan or done as emergencies due to the losses?

I'm not sure the summary of other Med sites is all that useful given the dominance of infrastructure. 

Author Response

We thank Reviewer #1 for his helpful comments.

Our notes on the relevant changes follow:

Was hydrodynamic or wave modeling done?

No hydrodynamic or wave modeling were done. However, wave data was taken into account. Please see section 2.1 (lines 89-106) in the revised manuscript.

line 87. The beach profiles probably vary greatly around the structures, but is there any estimate of a clousure dept?

We added the following paragraph in the end of section 2.1 (lines 107-113) in the revised manuscript.

Based on high-quality wave measurements recorded in Haifa and Ashdod Since 1992-1993, the closure depth (h) for the Israeli coast was calculated according to Birkemeier (1985) expression:

h = 1.57 He,

where He is the nearshore significant wave height (m) exceeded only for 12 hours per year. The calculated He for the Israeli coast, except Haifa Bay, is 5.1 to 5.8 m which means the closure depth is between 80. and 9.1 m. (Bitan and Zviely, 2019). In Haifa Bay however, the calculated He is 3.0 to 3.3 m which means the closure depth is between 4.7 and 5.2 m (Zviely et al., 2009).

line 95. What is the rate of LST? 

Based on wave data measured in Haifa and Ashdod between 1994 and 1997, Perlin and Kit (1999) estimated the wave-induced net LST rate along the Israeli coast. The annually average rate decrease form 400,000 m³ in Ashkelon (south Israel) to 200,000 m³ in Ashdod, 100,000 m³ in Tel Aviv, and only 70,000 m³ in the Carmel coast (just before the longshore current bypass the Carmel headland and penetrated in to Haifa Bay), net LST to the north.
Current study conducted by Zviely and Kit (in preparation), updated the rates for the Israeli coast, based on long-term wave data (1992-2017) measured in Haifa and Ashdod. According this study, the annually average LST rate decrease form 300,000 m³ in Ashkelon to 100,000 m³ in Ashdod, net LST to the north.

Any evidence of divergence along the coast? 

The wave-induced and wind-induced longshore currents and the sand transported by them are active in both directions. However, the long-term net longshore sand transport (LST) runs northward along the entire coast, up to Haifa Bay, the final depositional sink of the Nile littoral cell (Carmel et al., 1985; Perlin and Kit, 1999; Zviely et al., 2007). This northward net LST results mainly from larger size of waves approaching from west-southwest and southwest compared to their counterparts from west-northwest and northwest. As a result of this net LST direction, sand accretion has developed at the southern side (up-stream), while the northern side (down-stream) of marine structures along Israel’s southern coast has eroded. Along Israel’s central and northern coasts, however, this morphological phenomenon is less dominant - even inverted around some small coastal structures. This apparent contradiction led some researchers to the mistaken claim that net LST north of Netanya runs southward (Emery and Neev, 1960; Goldsmith and Golik, 1980, Shoshany et al., 1996; Golik, 1993; 1997; 2002). Such claims - if correct - would require a convergence zone between Tel-Aviv and Herzliya, where huge amounts would accumulate. Coastal and seabed observations in this region fail to detect any such accumulation (Zviely et al., 2000; Klein and Zviely, 2001; Dror, 2005).

Figure 3. Could cliff erosion be (or have been) the source of sand to the beach?

The contribution of coastal cliff erosion, as a source of sand to the beach, is limited and for a short time. The sediment that eroded form the cliff is mainly very-fine and wash rapidly offshore during winter season (i.e. wave storm).

line 160 and 282. were the repeated renourishments part of the original plan or done as emergencies due to the losses?

The repeated re-nourishments were not a part of the original plan, but as compensating the sand losses.

I'm not sure the summary of other Med sites is all that useful given the dominance of infrastructure.

Although the Med sites analogies is not perfect they can supply a sufficient factors for comparison. It was indicated in the introduction.

Reviewer 2 Report

Interesting paper and important for Israeli coast and beach maintenance

Methods and case studies well explained. Literature review and comparison OK. To improve scientific relevance some more analysis could be easily performed such as volume balance of cross sections. See questions als added in the draft text (attached) and below.

describe applied dredging technology shortly: Trailing Suction Hopper discharging via floating pipeline? spraying / rainbowing on the beach / nearshore?

Explain design: what was the computed yearly LST and what was the sand volume loss = .... m3? Does it match with the observed sand loss over the considered time?

is the sand transported off-shore to deeper water where it cannot recirculate to the coast again by waves and tide? Is the coastal slope that steep? What is the natural yearly supply of sand versus erosion by winter storms in the cross section? Therefore gentle slopes but coarse sand required. compare volumes with annual LST. is there a deep-sea sink or a circulation in the surf zone? Role of dunes as a sand supplier (or lack of dunes)?

Cross section of coast profile available? Average slope = 1:75 not very gentle. cross sections: before / after nourishment and after storm? Note a wide and gently sloping beach wil mitigate wave effects 

that is really very fine sand easily suspended by waves or currents

Perhaps recent detailed sattelite image analysis could reveal effect of breakwater extension Haifa port? Longshore currents may be deflected?

Can you put the Israeli projects in the same format in the table 1 for comparison?

Author Response

We thank Reviewer #2 for his helpful comments.

Our notes on the relevant changes follow:

describe applied dredging technology shortly: Trailing Suction Hopper discharging via floating pipeline? spraying / rainbowing on the beach / nearshore?

In the revised manuscript we replaced paragraph 2.2.1. with paragraph 2.2.2., in order to describe the applied dredging and sand beach nourishment technology (Please see the end of paragraph 2.2.1.):

The sand was deposited between the coastline and water depth of 3 m by rainbowing via a discharge pipe at the bow of the dredging vessel (SIMI - operated by EDT Marine Construction) anchored at a water depth of 6 m. The dredging vessel conducted up to four cycles per day, about 800-1,600 m³ of sand per load.

Explain design: what was the computed yearly LST and what was the sand volume loss = .... m3? Does it match with the observed sand loss over the considered time?

For the design of all nourishment projects in Israel, no computed yearly LST was done. However, Perlin and Kit (1999) estimated the wave-induced net LST rate along the Israeli coast and suggested that the annually average rate decrease form 400,000 m³ in Ashkelon (south Israel) to 200,000 m³ in Ashdod, 100,000 m³ in Tel Aviv, and only 70,000 m³ in the Carmel coast (just before the longshore current bypass the Carmel headland and penetrated in to Haifa Bay), net LST to the north. Current study conducted by Zviely and Kit (in preparation), updated the rates for the Israeli coast, based on long-term wave data (1992-2017) measured in Haifa and Ashdod. According this study, the annually average LST rate decrease form 300,000 m³ in Ashkelon to 100,000 m³ in Ashdod, net LST to the north.
The computed yearly LST rates above match with the observed sand loss over the considered time in Ashdod, Ashkelon and Haifa Bay.

Is the sand transported off-shore to deeper water where it cannot recirculate to the coast again by waves and tide? Is the coastal slope that steep? What is the natural yearly supply of sand versus erosion by winter storms in the cross section? Therefore gentle slopes but coarse sand required.

No monitoring was done to study the cross-shore sand transport between the nourished areas and deeper water in all the projects. That is one of the our conclusions (see no. 5).

Compare volumes with annual LST. is there a deep-sea sink or a circulation in the surf zone? Role of dunes as a sand supplier (or lack of dunes)?

In general, the Israeli shallow continental shelf (i.e. 0-30 m deep) is moderate (1:50-1:80) and covered by fine sand. Further 3 km offshore until the continental edge (30-100 m deep), the shelf is flat (1:100), covered by silt and clay, and no deep-sea sink is existed.

Regarding circulation in the surf zone -- no monitoring was done to study the cross-shore sand transport between the nourished areas and deeper water in all the projects.  

No dunes are existed in all nourished sites.

Cross section of coast profile available? Average slope = 1:75 not very gentle. cross sections: before / after nourishment and after storm? Note a wide and gently sloping beach will mitigate wave effects that is really very fine sand easily suspended by waves or currents.

Unfortunately, no detailed morphological and sedimentological monitoring program was conducted before / after the nourishment, and after wave storm. That lack of data led to conclusion no. 5.

Perhaps recent detailed sattelite image analysis could reveal effect of breakwater extension Haifa port? Longshore currents may be deflected?

It is reasonable that the Haifa Port main breakwater extension (i.e. the "Bay Port Project") changed the wave regime in south Haifa Bay, that impact the local longshore current.

Base on this assumption, the Coastal and Marine Engineering Research Institute (CAMERI) on behalf of the Israel Ports Company (IPC), conducting now days a comprehensive study to better understand the impact of the main breakwater extension on the wave and current regime in south Haifa Bay. For this study, wave, hydrodynamic and sedimentological models are operate and bathymetric survey is plan every few months.

Can you put the Israeli projects in the same format in the table 1 for comparison?

We added the Israeli projects to table 1 (please see appendix)

Reviewer 3 Report

Logical structure. The article has numerous correct references to literature. It is based on numerous measurement data, which are properly interpreted and presented.

Author Response

We thank Reviewer #3 for his helpful comments.